# Role of Transcription Factor BEND3 and Its Potential Effect on Cancer Progression

**DOI:** 10.3390/cancers15143685

**Published:** 2023-07-20

**Authors:** Sarah Naiyer, Lalita Dwivedi, Nishant Singh, Swastik Phulera, Vijay Mohan, Mohammad Kamran

**Affiliations:** 1Department of Biomedical Science, University of Pennsylvania, Philadelphia, PA 19104, USA; 2Faculty of Science, Department of Biotechnology, Invertis University, Bareilly 243122, UP, India; 3Cell and Gene Therapy Division Absorption System, Exton, PA 19341, USA; 4Initium Therapeutics, 22 Strathmore Rd., STE 453, Natick, MA 01760, USA; 5Department of Biosciences, School of Basic and Applied Sciences, Galgotias University, Greater Noida 203201, UP, India; 6Department of Pathology and Laboratory Medicine, Weill Cornell Medicine, New York, NY 10065, USA

**Keywords:** BEND3, tumor suppressor, oncogenic driver, cancer therapy

## Abstract

**Simple Summary:**

Cancer is the leading cause of death worldwide and it is estimated that approximately one in six deaths globally are due to cancer. Understanding the causes of cancer is critical for prevention, early detection, treatment optimization, public health planning, and to make informed decisions to reduce the burden of cancer on individuals and society. Transcription factors via multiple pathways have been shown to exert significant influence over gene expression programs and cellular processes involved in cancer progression. Understanding the role of transcription factors during these processes can provide critical insights into the underlying molecular mechanism and help develop therapeutic interventions. This review summarizes the possible pathways via which the transcription factor BEND3 may serve as a driver for cancer development.

**Abstract:**

BEND3 is a transcription factor that plays a critical role in the regulation of gene expression in mammals. While there is limited research on the role of BEND3 as a tumor suppressor or an oncogene and its potential role in cancer therapy is still emerging, several studies suggest that it may be involved in both the processes. Its interaction and regulation with multiple other factors via p21 have already been reported to play a significant role in cancer development, which serves as an indication of its potential role in oncogenesis. Its interaction with chromatin modifiers such as NuRD and NoRC and its role in the recruitment of polycomb repressive complex 2 (PRC2) are some of the additional events indicative of its potential role in cancer development. Moreover, a few recent studies indicate BEND3 as a potential target for cancer therapy. Since the specific mechanisms by which BEND3 may contribute to cancer progression are not yet fully elucidated, in this review, we have discussed the possible pathways BEND3 may take to serve as an oncogenic driver or suppressor.

## 1. Introduction

Cancer is a complex and multifaceted disease characterized by the uncontrolled growth and division of abnormal cells within the body. This rapid cell proliferation may give rise to benign and malignant tumors. While benign tumors are non-cancerous, malignant tumors invade nearby tissues and spread to other body parts. There is increasing evidence to suggest that tumors contain a small population of cancer stem cells (CSCs), which share similar self-renewal and differentiation properties as embryonic stem cells (ESCs) [1,2]. CSCs divide rapidly and are thought to help maintain the tumor’s growth and microenvironment. ESCs have the capacity to become nearly any cell type and the signals that prompt stem cells to switch off pluripotency and commit to their final functional lineage are still ambiguous.

Transcription factors and cell signaling pathways play critical roles in maintaining cellular functions and homeostasis. They often work in sophisticated networks and can interact with one another, forming complexes involved in different aspects of gene regulation, cell differentiation and development, cell cycle control, response to environmental stimuli, immune response, and pathogenesis. Aberrant expression or mutation in transcription factors can contribute to the development and progression of tumors. Overexpression of oncogenic transcription factors like MYC and NF-kB is shown to be associated with abnormal cell signaling and epigenetic regulations causing many cancer types and is linked to increased cell proliferation and survival [3,4]. On the other hand, tumor suppressor transcription factors like p53, BRCA1, and FOXO3a regulate normal cellular functions by modulating the expression of genes involved in DNA repair, cell cycle arrest, and apoptosis, and thus prevent cancer development. Overall, the delicate interplay between transcription factors, cell signaling, and gene expression is critical to tumor development and growth.

BANP, E5R, and Nac1 (BEN)-Domain containing protein 3 (BEND3), a transcription factor, has been recently identified to regulate pluripotency by repressing pro-differentiation genes both in humans [5] and mice [6]. It is highly expressed in pluripotent cells and binds to promoters of genes involved in differentiation, regulating the transcription of these genes. It is known to interact with various chromatin modifiers and transcriptional repressors such as the NoRC complex [7], HDACs [8,9], PICH1 [10], Sall4 [9], and with members of the nucleosome remodeling and deacetylase (NuRD) complex in a context-dependent manner [8,11]. BEND3 occupies enhancers of CGI-associated genes in mouse embryos, regulating their transcription [6]. In a recent study on the genome-wide CRISPER/Cas9 knockout screen in acute myeloid leukemia (AML) cells, BEND3 knockout showed resistance to TAK-243, an inhibitor of ubiquitin-like modifier-activating enzyme 1. TAK-243 is a phase I clinical trial drug for advanced malignancies [12]. Elevated BEND3 expression is also reported in multiple cancer types across The Cancer Genomic Atlas (TCGA) Program [13]. Although the role of BEND3 in cancer progression is still emerging and gaining prominence, nonetheless, the exact mechanism is not yet fully understood. The objective of this review is to provide a perspective on the understanding of the possible pathways BEND3 may participate in to play a role in tumorigenesis.

## 2. BEND3 Structure and Its Interaction with DNA

BEND3 is expressed from chromosome 6 in the human genome and has three introns and four exons. There are two reported transcriptional variants; nonetheless, both express the same functional protein. It has a long N-terminal loop followed by four BEN domains (BD1–4) of 80 amino acids each with distinct molecular interactions and functions. It is highly conserved across vertebrates. The nuclear localization signal (NLS) is present in the N-terminal loop. It forms an octameric higher-order structure [10]. Although the crystal structure of a full-length protein is not yet available, the structure of individual domains in mice (BD1, BD3, and BD4) [6] and humans (BD1 and BD4) [10,14] are available. The human BD1 domain is believed to be involved in the protein–protein interaction as it contains a coiled-coil domain and interacts with PICH1. The BD1 domain is exclusively composed of α helices with no beta turns [10]. The BD4 domain of BEND3 is composed of six α helices, two short helical turns, and two β strands. The DNA interaction region sits at the C-terminal and spans from α5 and α6. It is important to notice that both BD1 and BD4 domains share similar folds but perform distinct molecular functions. The BD1 domain lacks DNA binding activity [14]. The mouse BD3 domain shows similarity with the BD4 domain at the sequence as well as structural level, with few differences. The BD4 domain harbors DNA/chromatin binding activity. Unlike humans, mouse BD4 comprises five α helices, with the DNA binding module formed by α1 to α4 and the hinge region between α4 and α5 mediating dimer formation to accommodate two independent DNA molecules [14,15]. The NLS of this 95 KDa human protein consists of a Lysine-Arginine-Lysine motif [16] and, once in the nucleus, the BD4 domain can recognize and regulate its target genes. The BD4 domain is unique among other BEN domains as it has six α helices instead of five loops in others. In addition, it binds DNA through α5–loop–α6 sites while others bind through α5 and loop between α3 and α4 [14]. Zhang et al. has elaborately explained the crystal structure of the BD4 domain spanning from 715 to 828 in association with DNA [6].

## 3. BEND3-Mediated Chromatin Regulation

BEND3 interacts with various chromatin modifiers, including the subunits of the NuRD complex [8], suggesting its role in global gene regulation. The NuRD complex modulates the chromatin structure at the bivalent and poised rRNA genes [17] and various other genes in the ESCs [6,18]. It plays significant roles in processes like neural stem cell reprogramming into iPSC [19], pericentric heterochromatin formation [20], the maintenance of pluripotency [17,21,22,23], transcription repression [24,25], and S-phase progression [8]. The NuRD complex comprises various subunits such as CHD3/4, HDAC1/2, MBD2/3, MTA1/2/3, and retinoblastoma-binding protein (RBBP4/7). Other subunits such as LSD1 and GATAD2A/B have also been reported to associate with this complex in certain types of cells. Several subunits of the NuRD complex have implications in cancer development and progression. Its components, including MTA1, HDACs, Sal4, and CHD4, are expressed in different cancer types and corroborate the tumor progression and poor prognosis [11,26]. MTA1 is reported to be overexpressed in a wide range of cancer types and this overexpression correlates with tumor grade, poor prognosis, and invasion status of the tumor [11]. MTA1 acts as a downstream effector molecule of the Myc oncogene [27]. The NuRD complex is reported to interact with various oncogenic transcription factors and mediate the transcription repression of many target genes. In diffuse large B cell lymphoma (DLBCL), MTA3 associates with an oncogenic transcription repressor, BCL6 [28], which plays a significant role in the development of a significant proportion of DLBCL [29]. BCl6 requires MTA3 to transcriptionally repress normal plasma cell differentiation [28]. MTA proteins also interact with the transcriptional repressor BCL11B in leukemia and lymphoma cell lines. BCL11B has an indispensable role in early T cell development [11]. In breast cancer cells, a transcription factor, TWIST, causes the recruitment of MTA2 containing the NuRD complex at the CDH1 (E-Cadherin) promoter so as to repress E-Cadherin and promote the epithelial to mesenchymal transition (EMT), which is a hallmark of cancer, while MTA3 of the NuRD complex has been reported to cause transcription repression of SNAIL, thus inhibiting EMT in breast cancer cells. Thus, the complex can act as either a promoter or inhibitor of EMT depending upon the context of cells [11]. An oncogenic chimeric protein, PML-RARα, also recruits the NuRD complex, which in turn recruit other epigenetic regulators to cause transcriptional repression, leading to the impairment of cellular differentiation in human acute promyelocytic leukemias [30]. It is speculated that transcription factors recruit this complex to the promoters of the genes implicated in cancer [26]. Nine subunits of the NuRD complex are reported to be upregulated in hepatocellular carcinoma [26]. LSD1, part of the complex, regulates metastasis in breast carcinoma and is found to be under-expressed in breast cancers [31].

HDACs are important players in tumorigenesis and tumor progression and are overexpressed in several different kinds of malignancies including breast, colorectal, liver, pancreatic, ovarian, cervical, prostate, renal, bladder, melanoma, and certain blood cancers [32]. The BEN domain might act as an adapter in the process of chromatin modification by HDACs [15]. However, MTA1 and HDACs of the NuRD complex represent potential therapeutic targets for cancer chemoprevention. BEND3 also interacts with another NuRD complex transcription factor, Sal4, which is required to maintain the stemness and pluripotency of embryonic stem cells [9]. Sal4 expression is mis-regulated in various hematological as well as solid malignancies [33]. It plays a significant role in regulating various genes including apoptotic genes; cell-surface marker (EpCAM); EMT-related genes such as TWIST1, SNAI1, VIM, ZEB, E-Cadherin, etc.; and epigenetic modifiers such as DNMT1 and LSD1 [33]. Thus, it can be speculated that via its interaction with these subunits, BEND3 could aid in tumorigenesis mediated by the NuRD complex. Interestingly, BEND3 showed weak interaction with HDAC2 in our docking screen. The docking of BD4 was performed with HADDOCK 2.4 software and results are shown in Figure 1. We found a significant interaction between the two proteins, though it was not very strong with a Z-score “−0.8”, and the RMSD value from the overall energy structure was 18.9 ± 0.3 [34,35].

BEND3 also interacts and stabilizes the nuclear remodeling complex (NoRC) by inhibiting the ubiquitination of TTF-1-interacting protein 5 (Tip5). BEND3 has two SUMOylation sites at K20 and K512 which are essential for NoRC stability. SUMOylated BEND3 interacts with the ubiquitin-specific protease 21 (USP21) deubiquitinase and prevents the ubiquitination of Tip5. Tip5 helps in the recruitment of this complex to the rDNA promoters via the TTF1 transcription factor. In the SUMOylation-deficient mutant of BEND3, USP21 was found to be destabilized, indicating the importance of BEND3 SUMOylation for its interaction with USP21. In the absence of BEND3, the interaction of Tip5 with the rDNA promoter along with the methylation of rDNA was severely decreased. Most of the rDNA repeats were kept epigenetically silent by the NoRC through the recruitment of DNMTs and histone-modifying enzymes [7]. Being a highly repetitive, heavily methylated, and actively transcribed region, the rDNA locus is very prone to genomic instability, which could ultimately lead to cancer development [36]. The rDNA locus is also known to exhibit notable contraction and expansion followed by the unequal exchange of sister chromatids [36]. Repeated expansion of this locus has been detected in colon and lung cancers [37], and may result in increased ribosome production in tumor cells [38]. Myc oncogene also increases rRNA synthesis by increasing the expression of RNA polymerase I, along with other transcription factors [36]. Several cancers show hypomethylation of the promoter region, while hypermethylation was observed in 28S and 5.8S rRNA coding regions and some spacer regions [36]. The loss of BEND3 results in increased H3K4me3 and decreased DNA methylation at the rDNA gene promoters with an increase in the levels of pre-rRNA, whereas its overexpression results in decreased H3K4me3 and H4Ac (pan) and increased H4K20me3 and H3K27me3 at rDNA promoters [7].

## 4. Epigenetic Regulation by BEND3

The study of heritable and reversible alterations that control gene expression without any change in the genome sequence refers to epigenetics. The dysregulation of epigenetic modifiers has been reported in many cancer types. The most significant epigenetic changes that occur during tumorigenesis include hypermethylation at the promoters of tumor-suppressor genes, global hypomethylation, and the alteration of histone marks. Increased expression levels of DNA methyl transferases (DNMTs) and histone deacetylases (HDACs) are commonly found in various cancer types. Histone methylases (HMTs) and demethylases (HDMs) also play important roles in cancer development. Histone methyl transferases such as mixed-lineage lymphoma (MLL1) and enhancer of zeste 2 (EZH2) and histone demethylases such as lysin-specific demethylase (LSD1) show elevated expression in various cancer types [39]. Thus, mutations or alterations in writers, readers, and erasers of epigenetic marks along with different members of chromatin regulation result in cancer development [40].

BEND3 is reported to mediate a switch in the chromatin state from constitutive heterochromatin to facultative heterochromatin during embryonic development and cancer [7,41]. DNA methylation and H3K9me3 mark the chromatin as constitutive heterochromatin, while DNA hypomethylation and H3K27me3 indicate the facultative heterochromatin regions. These two marks are mutually exclusive and do not exist simultaneously at a particular locus. One of these states is adapted by the pericentromeric region during development and cancer. DNA methylation inhibits the recruitment of polycomb group proteins (EZH2) and regulates the chromatin architecture to some extent. BEND3 is abundantly present at the pericentromeric regions and helps the polycomb group to be recruited at that locus. This recruitment leads to the accumulation of H3K27me3 marks in the absence of DNA methylation and H3K9me3, thus regulating the constitutive to facultative heterochromatin switch [41]. Pericentromeric regions in various cancer cell lines exhibit the recruitment of polycomb group proteins to a large extent and constitute characteristic structures known as polycomb bodies [42,43].

Saksouk et al. identified BEND3 as a methylation-regulating protein that acts as an important player in the recruitment of PRC2 to major satellites. This recruitment occurs under specific conditions such as the fertilization or impairment of DNA methylation and H3K9me3 marks, as reported in various cancer types [41]. BEND3 recruits the MBD3/NuRD complex, which plays an important role in the further recruitment of PRC2 to certain CpG islands in embryonic stem cells [44]. In transformed cell lines, the exogenously expressed BEND3 protein was found to interact with the heterochromatin region [9]. These data suggest that BEND3 functionally interacts with the PRC2 complex. As PRC2 has a role in differentiation and development, it can be suggested that BEND3 plays an important role in these physiological processes. Furthermore, the PRC2 complex is also implicated in the development and progression of different types of cancer [45,46]. It is a possibility that BEND3 could also exhibit cancer-promoting effects and act together with this complex in the modulation of epigenetic marks at various loci to promote cancer development.

## 5. BEND3 in Cell-Cycle Regulation

BEND3 downregulates p21 [5], which inhibits cell-cycle progression at the G1 or G2 phase, implying an increase in the cell-cycle rate in the presence of BEND3. We performed a simulation of the interaction profile of the BD4 domain with p21, as shown in Figure 2. Surprisingly, our docking result showed a strong interaction between the above-mentioned proteins with a Z-score of “−1.4” and the RMSD value from the overall lowest energy structure was 0.9 ± 0.6 [34,35]. This strongly supports BEND3’s role as an oncogene and is a matter of further exploration. On the contrary, the overexpression of BEND3 is associated with extensive heterochromatin formation and the premature condensation of chromatin, leading to cell-cycle arrest in the early S1 phase [9]. Would this call for BEND3 to be a tumor suppressor? This Yin Yang nature of BEND3 on cell-cycle progression suggests that it acts in a context-dependent manner and is indeed a subject for further investigation.

The PICH (Polo-like kinase 1-interacting checkpoint helicase; also known as ERCC6L) is an ATP-dependent DNA translocase which resolves ultra-fine anaphase bridges (UFBs) in humans. The PICH protein acts as a prime player in recognizing topological stress through its co-operation with topoisomerase II [10] during anaphase. This stabilizes DNA in stress conditions by means of resolving the entangled sister chromatids, probably formed following various topological changes during replication. If this entanglement is not resolved, chromatin bridges or UFBs are formed during the anaphase of mitosis, resulting in the mis-segregation of sister chromatids [47]. PICH acts as a safeguard of chromosome integrity/stability and could act as a tumor-suppressor gene by maintaining genome stability. PICH has been found to co-localize with other repair proteins such as BLM (Bloom syndrome helicase) on UFBs during mitosis. It covers the UFBs all along their length and helps recruit other DNA repair complexes to this site. On the other hand, the expression of PICH is reported to be increased in simple as well as triple-negative breast cancer (TNBC), and it is found to be essential for TNBC. Tumors with a high expression of PICH exhibited increased chances of metastasis and decreased survival. Cell viability and proliferation of TNBC was observed to be decreased after the loss of PICH, and its downregulation resulted in the induction of apoptosis and impairment in mitosis. Moreover, the silencing of PICH also caused a decrease in the tumor growth of TNBC under in vivo conditions [48]. These studies project PICH as an oncogene.

The association of the BEND3 protein with PICH at UFBs may indicate yet another pathway used by BEND3 towards cancer progression. PICH contains two TPR domains that take part in protein–protein interactions. PICH and BEND3 interact with each other directly. This interaction is mediated by the TPR domain at the N-terminal of PICH and the first BEN domain (BD1) of BEND3, and Asp265 is an important residue for this interaction. The mutation in this residue can hinder the function of PICH, resulting in chromosome instability. Therefore, it can be speculated that BEND3 and PICH might act together to inhibit the formation of UFBs [10,49] and, thus, act to safeguard the genome’s integrity. Any breach in the genome integrity predisposes the cells to malignant transformation [50]. Hence, by way of guarding genome integrity, it is likely that PICH1 and BEND3 inhibit the malignant transformation of cells, resulting in decreased tumorigenesis. The association and colocalization of BEND3 with telomeric repeat binding factor 2 (TRF2) further strengthen its possible role in genomic integrity and cell-cycle regulation.

## 6. Role of BEND3 in Pluripotency Maintenance

Gene expression and ChIP Seq data show that high BEND3 expression plays a role in pluripotency maintenance [5,6] in ESCs, and interacts with the promoters of various differentiation-associated genes [5]. BEND3 knockdown results in the induction of differentiation in human embryonic carcinoma cells, NTERA2. This depletion results in the downregulation of certain genes involved in DNA replication, chromosome condensation, cell-cycle progression, and the upregulation of genes involved in cell-to-cell interaction. Loss of BEND3 also results in altered levels of genes involved in neuronal differentiation. This loss is accompanied by changes in gene expression involved in MAPK and p53 pathways, as exemplified by the increase in the levels of BDNF and TGFβ2 genes. These two proteins not only have roles in brain development, but also control brain signaling from synaptic regulation to neurodegenerative disorders [51,52,53,54,55]. They have been shown to be involved in cancer development and progression [56,57]. BEND3 represses Calreticulin, which is involved in brain function [58,59,60] and also inhibits retinoic acid expression, which is essential for neuronal differentiation [61].

BEND3 binds at the G-quadruplex-rich promoters of differentiation-associated genes and cell-cycle regulators including p21 and subsequently represses their expression. Overexpression of BEND3 results in the increased H3K27me3 mark at the promoter of these genes, which is possibly mediated by the recruitment of the PRC2 complex by BEND3. This repressive mark results in the pausing of RNA pol II at these promoters, finally leading to the repressed transcription. The polycomb proteins play a critical role in embryonic development and cell lineage commitment [62]. p21 causes cell-cycle arrest, which precedes differentiation in the developing mouse embryos [63], and its overexpression is known to induce differentiation in normal as well as tumor cells. p21 is essential for the survival of differentiating neuroblastoma cells. Thus, depending upon the context, p21 can positively and negatively influence the differentiation process. A decrease in the expression of p21 by BEND3 enables the cells to be in an active state. The overexpression of BEND3 results in the downregulation of these differentiation-associated genes, but this overexpression does not display any change in the expression of proliferation-associated genes [5].

BEND3 binds at its own promoter and autoregulates its expression [5,14]. The absence of BEND3 was reported to be lethal between embryonic stage E3.5 and E6.5. The ESCs were observed to be smaller and shrunken in the BEND3 knockouts (KOs), further pointing towards its role in the differentiation process. In these KOs, the bivalent genes were upregulated, suggesting that BEND3 preferentially decreases the expression of bivalent genes during the process of differentiation [6]. ESCs and other pluripotent cells divide rapidly and have the capacity to become nearly any cell type in the body. The signals that prompt stem cells to switch off pluripotency and adopt their final functional state are still unclear. BEND3 shuts down the expression of hundreds of genes associated with differentiation, maintaining the cell’s stem-cell-like status, thus regulating pluripotency.

## 7. Discussion and Future Prospects

The expression of BEND3 is relatively much higher in transformed or cancerous cell lines including human U2OS, HeLa, and mouse NIH3T3 [9]. BEND3 was also found to be upregulated in metastatic breast tumors [64]. The eXplainable Artificial Intelligence (XAI [65] and CRISPR screen [12] identified BEND3 as an important molecule in breast cancer and leukemia, further strengthening its role in cancer progression. In a genome-wide CRISPR screen, BEND3 was identified as an effector molecule for TAK-243 sensitivity in AML. TAK-243 is a first-in-class inhibitor of the ubiquitin-like modifier-activating enzyme 1 (UBA1) [50], which is essential for the initiation of the ubiquitin conjugation cascade [66]. TAK-243 is undergoing phase I clinical trials for the treatment of various advanced cancers. The knockout of BEND3 resulted in the dampening of TAK243-mediated effects for instance proteotoxic stress, ubiquitylation inhibition, and DNA-damage response. This effect on the sensitivity of the drug was mediated by the upregulation of the ABCG2 transporter, breast-cancer-resistance protein (BCRP), in various cancer cell lines, indicating that BEND3 regulates the expression of BCRP [12]. By inhibiting BCRP, BEND3 sensitizes cancer cells to the drug by means of inhibiting its transport out of the cell. Interestingly, BEND3 is located in a region that is frequently found deleted in leukemias, lymphomas, and carcinomas of the breast, prostate, and ovary in humans [67,68,69,70,71]. A possible tumor suppressor, HACE1 has been reported at this locus [72]. These observations strongly suggest that BEND3 may play an important role in cancer development and progression, also supported by a recent study which reports higher BEND3 expression in cancer cells across TCGA database compared with normal cells [13].

BEND3 associates with poised bivalent promoters and enhancers and controls the expression of downstream genes [9]. The downregulation of BEND3 leads to differentiation and, eventually, cessation of the proliferation. It binds and co-ordinates with many other molecular regulators of cell-differentiation pathways [7]. Notably, BEND3 interacts with multiple proteins and protein complexes which are associated with cancer development and progression, including HDAC2, HDAC3, Sall4, PRC2, NuRD, and NoRC. Many proteins of the PRC2 complex and epigenetic modifiers have been identified to play roles in cancer progression [73]. In a recent study, Abid et al. compared the expression of PRC2 complex protein EZH2 and BEND3 across all the reported cancers in TCGA database and established a very strong correlation between the mRNA expression of these two. The correlation was even greater in breast cancer and in glioblastoma [13]. Furthermore, it also causes transcriptional repression of p21, a cell-cycle checkpoint protein, and arrests the cell cycle in the G1 or G2 phase, suggesting its possible oncogenic role. Stem cells have the capacity to repopulate a tumor after it has shrunk during treatment. Finding a molecular switch that will shift cancer cells away from proliferation and toward differentiation could aid in cancer treatment. Its expression is critical in determining a cell’s fate, making it an attractive target for potential medical interventions. All these studies indicate that BEND3 works in a context-dependent manner, and further work is warranted in this direction.

## 8. Conclusions

Cancer is a mortiferous disease which is characterized by a mass of cells undergoing fast and uncontrollable growth. These cells more often detach from their source of origin and move to other parts of the body. Various genetic and epigenetic changes and the accumulation of mutations over a period of time may result into the development of cancer. Changes such as promoter-specific hypermethylation, global hypomethylation, and various types of histone modifications also play a significant role in cancer development [74]. Understanding the role of transcription factors is key to identifying potential therapeutic targets and developing effective treatments. In this manuscript, we have tried to establish the link between the transcription factor BEND3 and cancer (Figure 3). We have discussed the recent advancements in the field and believe this article will provide the base and motivation for further research.

## Figures and Tables

**Figure 1 cancers-15-03685-f001:**
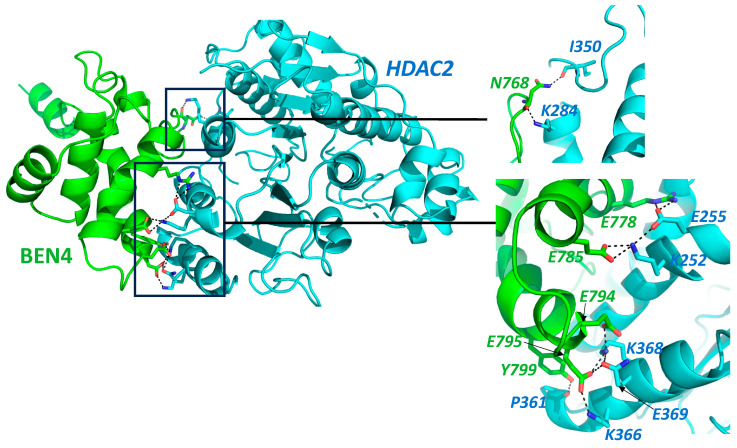
Protein–protein interaction prediction through HADDOCK 2.4 Docking software [34,35]. Interaction of BEN4 domain of BEND3 (Green) with HDAC2 (Cyan), Full protein structure along with interaction interface, the interaction area is zoomed in the insets. Protein structure with highlighted interacting amino acid. Interactions are shown by dotted lines. Amino acid labels color coded to match the contributing partner. Main chain shown only in cases it is involved in the proposed interaction.

**Figure 2 cancers-15-03685-f002:**
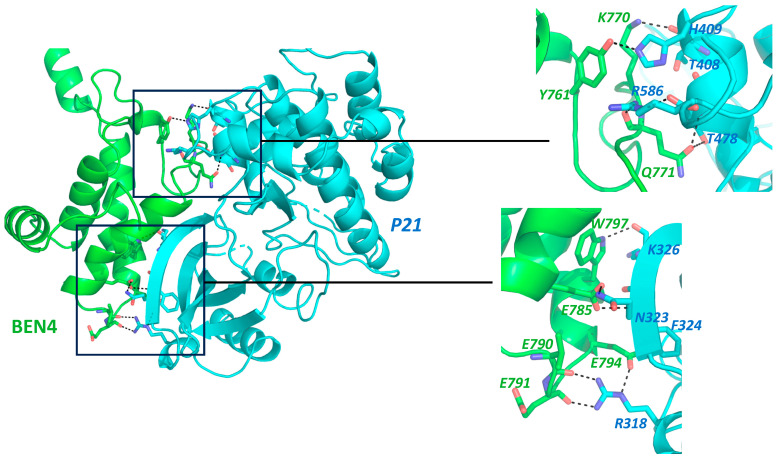
Protein–Protein interaction prediction through HADDOCK 2.4 Docking software [34,35]. Interaction of BEN4 domain of BEND3 (Green) with p21 (Cyan), Full protein structure along with interaction interface, Insets: Protein structure with highlighted Interacting amino acid. Interactions were shown by dotted lines. Labelling color corresponds to the contributing partner. Main chain shown only in cases it is involved in the proposed interaction.

**Figure 3 cancers-15-03685-f003:**
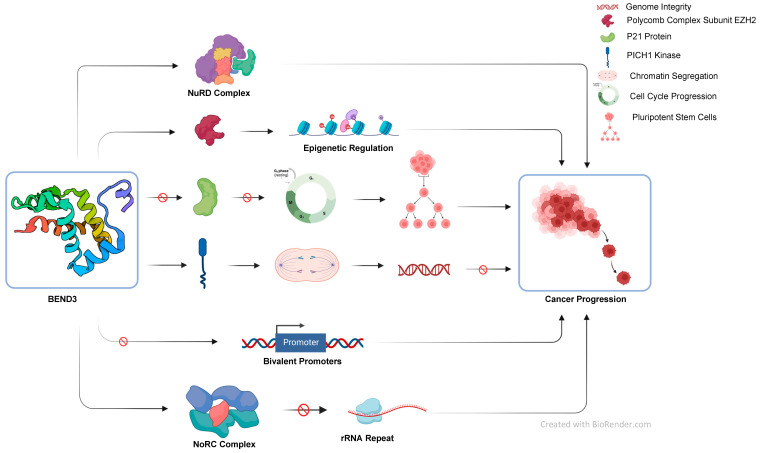
Schematic summarizing the role of BEND3 protein studies so far in various cell processes.

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
