# Peer review of "Role of Transcription Factor BEND3 and Its Potential Effect on Cancer Progression"

_cancers, 2023, doi:10.3390/cancers15143685_

Round 1

Reviewer 1 Report

Some focus and description of various tools and methods used for this literature review would enhance the credibility of this review .

Author Response

Comment: Some focus and description of various tools and methods used for this literature review would enhance the credibility of this review

Response: We have added that in the acknowledgment now. Thanks!

Reviewer 2 Report

In this manuscript, Naiyer et al., summarize a wealth of published findings related to the role of the TF BEND3 with emphasis on cancer. The manuscript provides information from diverse themes and cellular phenomena ranging from structural features of BEND3 to pluripotency. This is important in order to highlight the critical commitment of a TF in vital cellular processes. On the other hand, the manuscript should be revised in order to achieve a comprehensive description of all these aspects. My opinion is that the authors should expand each section and provide more details for the mechanisms that they describe. In addition, they have to re-write several parts, since in many sentences same words are repeated, and that makes the text not attractive. Finally, considering that the title is clearly relevant to the function of the TF in cancer progression, the authors have to describe more functional evidence regarding BEND3 commitment.

 Several points are outlined below:

Introduction 

[1] “Cancer is characterized by the uncontrolled growth and spread of abnormal cells in the body. The cells begin to grow and divide uncontrollably, forming a mass of cells called a tumor”

These two sentences are too naïve to introduce the review and in addition are redundant at a significant scale. The authors have to rephrase.

[2] “Though the exact mechanism through which BEND3 contributes to cancer development and progression is not yet fully understood, there are studies indicating it as a potential target for cancer therapy. The objective of this review is to summarize the possible pathways through which BEND3 is speculated to play a role as an oncogene”

"through which" is repeated

[3] “It is 828 amino acids long protein and expressed from chromosome number 6.”

rephrase

Main text

[4] “BEND3 structure and its interaction with DNA”

The authors give a lot of space for describing in detail the structural data of BEND3. Given that the mission of the review is not structural but functional with emphasis on cancer progression, I suggest reduction of this section.

The authors have to re-write several parts, since in many sentences same words are repeated, and that makes the text not attractive. 

Author Response

Comment: In this manuscript, Naiyer et al., summarize a wealth of published findings related to the role of the TF BEND3 with emphasis on cancer. The manuscript provides information from diverse themes and cellular phenomena ranging from structural features of BEND3 to pluripotency. This is important in order to highlight the critical commitment of a TF in vital cellular processes. On the other hand, the manuscript should be revised in order to achieve a comprehensive description of all these aspects. My opinion is that the authors should expand each section and provide more details for the mechanisms that they describe. In addition, they have to re-write several parts, since in many sentences the same words are repeated, and that makes the text not attractive. Finally, considering that the title is clearly relevant to the function of the TF in cancer progression, the authors have to describe more functional evidence regarding BEND3 commitment.

Response: We have revised the manuscript and expanded each section to provide more mechanistic details. We have also taken care of the repetitive words and re-written them. We have also described more functional evidence regarding BEND3 commitment both in the main text and the discussion.

Several points are outlined below:

Introduction 

[1] “Cancer is characterized by the uncontrolled growth and spread of abnormal cells in the body. The cells begin to grow and divide uncontrollably, forming a mass of cells called a tumor”

These two sentences are too naïve to introduce the review and in addition are redundant at a significant scale. The authors have to rephrase.

Response: We have rephrased the introduction.

[2] “Though the exact mechanism through which BEND3 contributes to cancer development and progression is not yet fully understood, there are studies indicating it as a potential target for cancer therapy. The objective of this review is to summarize the possible pathways through which BEND3 is speculated to play a role as an oncogene”

"through which" is repeated

Response: We have changed it.

[3] “It is 828 amino acids long protein and expressed from chromosome number 6.”

Rephrase

We have removed this line.

Main text

[4] “BEND3 structure and its interaction with DNA”

The authors give a lot of space for describing in detail the structural data of BEND3. Given that the mission of the review is not structural but functional with emphasis on cancer progression, I suggest a reduction of this section.

Response: We have reduced this section now.

The authors have to re-write several parts, since in many sentences same words are repeated, and that makes the text not attractive. 

Response: We have re-written the parts to remove repetition.

Reviewer 3 Report

The authors have done a great job summarizing what is known about BEND3, a relatively unknown protein.  They have documented a number of factors that BEND3 can be involved with and have given sufficient information to peak the interest of readers. A few suggestions:

1. What is the influence of BEND2 on apoptosis?

2. What is the influence of BEND2 on metabolism?

3. They state on line 146 " identified BEND3 as a methylation-sensitive protein"...what does this mean? Have the methylation sites of BEND3 been identified? How many CpG sites are there on the promoter region of BEND3?

4. What is the DNA structure of BEND3 intron/exons? Protein domain structure? Phosphorylation sites if any? 

5.  A number of studies suggest a role in brain signaling pathways. A description of this should be made in this review as it appears to be  omitted

6. Lastly, the authors should check for any relevant reviews on BEND3 including 2022 and 2023 reviews. I located one....https://pubmed.ncbi.nlm.nih.gov/36648610/ IN 2023

If these questions can be addressed I would be willing to read a revised version of this manuscript. 

Author Response

The authors have done a great job summarizing what is known about BEND3, a relatively unknown protein.  They have documented a number of factors that BEND3 can be involved with and have given sufficient information to peak the interest of readers. A few suggestions:

1. What is the influence of BEND2 on apoptosis?

Response: It will be very interesting to study the role of BEND3 protein in apoptosis, nonetheless, there is no direct role of BEND3 protein  in apoptosis yet reported.

2. What is the influence of BEND2 on metabolism?

Response: No work has been reported to show a direct link between BEND3 protein and metabolism.

3. They state on line 146 " identified BEND3 as a methylation-sensitive protein"...what does this mean? Have the methylation sites of BEND3 been identified? How many CpG sites are there on the promoter region of BEND3?

Response: We apologize for the confusion. We have changed the word sensitive to regulating protein. BEND3 controls the methylation of the downstream protein where it is recruited. It is a part of the NURD complex which methylates the DNA and hence represses the expression of the downstream genes. In the absence of BEND3, the NURD complex loses its activity. There is no reported post translation methylation modification site in BEND3 protein, though it gets sumoylated which is explained in the main text. (ABID et al PNAS 2015). There is one reported CpG site at the promoter of BEND3. (UCSC genome browser)

4. What is the DNA structure of BEND3 intron/exons? Protein domain structure? Phosphorylation sites if any? 

 Response: According to NCBI database there are two transcript variants of BEND3, nonetheless, there are no reported translated variants of the protein. BEND3 has four exons and three introns (both variants) and we have now included this information in the review. As discussed in the main text “Structurally, BEND3 contains four protein domains (BD1-BD4) with distinct molecular interactions and functions. There is no reported phosphorylation site for BEND3 yet.

4. A number of studies suggest a role in brain signaling pathways. A description of this should be made in this review as it appears to be  omitted

Response: Since these studies do not have a direct implication on the role of BEND3 in cancer progression we initially omitted these. But we have not included some of these studies in the main text of the review. 

5. Lastly, the authors should check for any relevant reviews on BEND3 including 2022 and 2023 reviews. I located one....https://pubmed.ncbi.nlm.nih.gov/36648610/ IN 2023

Response: We were not able to access this particular article but have now cited other recent literature.